# Intraoperative Neurovascular Bundle Preservation with Hyaluronic Acid during Radical Brachytherapy for Localized Prostate Cancer: Technique and MicroMosfet In Vivo Dosimetry

**DOI:** 10.3390/biomedicines10050959

**Published:** 2022-04-21

**Authors:** Pedro J. Prada, María Ferri, Juan Cardenal, Ana García Blanco, Elisabeth Arrojo, Javier Anchuelo, Ivan Diaz de Cerio, Pedro Lastra, Alejandro Fernández, Andrés Vázquez, Samuel Ruiz Arrebola

**Affiliations:** 1Radiation Oncology Department, Hospital Universitario Marqués de Valdecilla, Cantabria, 39008 Santander, Spain; pjprada@pjprada.com (P.J.P.); juan.cardenal@scsalud.es (J.C.); anasoledad.garcia@scsalud.es (A.G.B.); elisabetestefania.arrojo@scsalud.es (E.A.); javiertomas.anchuelo@scsalud.es (J.A.); ivan.diazdecerio@scsalud.es (I.D.d.C.); 2Radiology Department, Hospital Universitario Marqués de Valdecilla, Cantabria, 39008 Santander, Spain; pedro.lastra@scsalud.es; 3Radiology Department, Hospital Sierrallana, Torrelavega, Cantabria, 39008 Santander, Spain; alejandro.fernandezf@scsalud.es; 4Radiation Physics Department, Hospital Universitario Marqués de Valdecilla, Cantabria, 39008 Santander, Spain; joseandres.vazquez@scsalud.es (A.V.); samuel.ruiz@scsalud.es (S.R.A.)

**Keywords:** brachytherapy, neurovascular bundle, preservation, technique

## Abstract

Purpose: To evaluate the reduction in the absorbed dose delivered to the neurovascular bundle (NB) in patients with localized prostate cancer treated with only HDR brachytherapy and NB protection with hyaluronic acid (HA) on the side of the prostate to increase the distance from NB to the radioactive sources. Methods: This is the first published report in the medical literature that studies a new approach to decrease neurovascular bundle toxicity and improve quality of life for patients with prostate cancer treated with radical brachytherapy as monotherapy. Transperineal HA injection on the side of the prostate into the lateral aspect of the prostate fat was used to consistently displace several autonomic fibers and vessels on the lateral wall of the prostate away from radiation sources. Results: When a protection in the form of an HA layer is placed, the reduction effect at the maximum dose is between 46% and 54% (calculated values), which means that the method for protection is highly recommended. The values of the absorbed dose calculated in this project have been compared with the ones given by the treatment planning system. Conclusions: This newly created space decreases absorbed dose in the NB, calculated with the TPS and measured by microMOSFET due to the thickness of HA.

## 1. Introduction

A variety of brachytherapy modalities are available for the radical treatment of localized prostate cancer. Low-dose-rate (LDR) and high-dose-rate (HDR) brachytherapy are accepted, with both treatments achieving similar rates of biochemical control, although with differing types of morbidities [1,2].

Radical brachytherapy is an accepted, effective, and safe therapy for localized prostate cancer. Contemporary series examining brachytherapy (BT) indicate excellent cancer-specific outcomes among younger patients [3,4,5]. Brachytherapy is as efficient as other interventions (prostatectomy, external beam radiotherapy) but less harmful and is especially attractive for patients with early prostate cancer [6,7,8,9].

The morbidity associated with brachytherapy treatment has decreased over the last decade as result of improvement in the brachytherapy technique. Young men with longer life expectancies are the population most likely to benefit from reduced normal tissue toxicity, particularly in terms of erectile preservation.

The sympathetic and parasympathetic nervous systems both play an important role in sexual function. Sacral parasympathetic fibers from S2–4 travelling through the pelvic plexus and forming the nervi erigentes dorsolateral to the bladder and prostate are responsible for the blood flow into the corpora cavernosa resulting in penile erection. Sympathetic fibers are responsible for the emission of semen from the seminal vesicles into the prostatic urethra and antegrade ejaculation. Thus, iatrogenic damage to the parasympathetic and sympathetic pathways during radical brachytherapy may result in erectile and ejaculatory dysfunction [10,11].

The purpose of the present study was to determine the reduction in the absorbed dose delivered to the neurovascular bundle (NB) using microMOSFET detectors (“in vivo” dosimetry) in patients with localized prostate cancer treated with only HDR brachytherapy and neurovascular bundle protection with hyaluronic acid (HA) on the side of the prostate to increase the distance from NB to the radioactive sources. With this technique, we believe many of the side effects can be either reduced or eliminated.

## 2. Material and Methods

This is the first published report in the medical literature that studies a new approach to decrease neurovascular bundle toxicity and improve quality of life for patients with prostate cancer treated with radical brachytherapy as monotherapy. Transperineal HA injection on the side of the prostate into the lateral aspect of the prostate fat was used to consistently displace several autonomic fibers (afferent and efferent) and vessels on the lateral wall of the prostate away from the effective and biologic life of radiation sources. In low- and low-intermediate-risk patients treated with HDR monotherapy, HA injection was administered before dose delivery of HDR. This study was approved by the institutional ethics committee, and all patients signed informed consent.

### 2.1. Study Model

The injection of HA was performed before needle placement, creating a minimum of a 1.5–2 cm space between the prostate capsule and neurovascular bundle. Having finished the HDR brachytherapy needle placement and before dose delivery, microMOSFET detectors were placed within the prostatic capsule and in neurovascular bundle. Intraoperative trans-rectal ultrasound (TRUS) was used for verification of the microMOSFET placement. In this fashion, we measured the delivered dose to both critical organs during the treatment “in vivo”.

Magnetic resonance pre- and post-implant imaging were used to assess the dimensions of the new space created. If necessary, magnetic resonance pre-implant fusion was performed with the intraoperative TRUS image.

Each patient received one implant and one fraction of HDR. The fraction dose was 22 Gy.

Brachytherapy procedures were conducted under spinal anesthesia. The needles were positioned by transperineal placement under real-time TRUS guidance using a template. Axial cross-sections were captured in 5 mm steps and transferred to the treatment planning software. The prostate gland, normal structures (urethra and rectum), and needle positions were identified and mapped based on the ultrasound image. Dose optimization was performed on the reconstructed applicator geometry using dose point and manual optimization algorithms to determine dwell positions and times.

The prostate without safety margins was then defined as the planning target volume (PTV) to be treated with the prescribed dose (PD). Based on the dose-volume histograms (DVH) data, the quality of plans and implants was evaluated using the following indicators:■The rectal dose was calculated at the anterior edge of the TRUS probe and was limited to ≤75% of the prescription dose.■The dose to any segment of the urethra was limited to ≤110% of the prescription dose. V120 and D100 of the prostatic urethra were determined (volume that received a dose of 120% and dose delivered to 100% of the urethra).■The PTV V90, V100, V150, and V200 (% of PTV receiving 90%, 100%, 150%, and 200% of the PD) were recorded.■D90 (dose delivered to 90% of the PTV) was calculated.

All patients were discharged from the center on the same day of the procedure between 6 and 8 h of implantation.

### 2.2. Technique of Hyaluronic Acid Injection

Based on our experience of rectal protection with HA in prostate cancer brachytherapy, the injection of HA was performed before needle placement [12,13]. We describe the technique below.

Step 1: The TRUS probe with the transperineal template was placed and fixed in the standard fashion. Magnetic resonance pre-implant images were fused with the intraoperative TRUS image. Then, all treatment needles were placed under TRUS guidance.

Step 2: Using TRUS guidance, an additional needle was placed, guiding the needle tip into the peri-prostatic fat, between the lateral prostate capsule (treated volume) and the neurovascular bundle, at the level of the maximum transverse diameter of the prostate (reference level). Under direct TRUS guidance, the needle tip was advanced to the level of the seminal vesicles and then to the level of the prostatic apex. Extreme care was taken not to perforate the prostate capsule with the needle tip.

Step 3: The additional needle was connected to the syringe containing HA. After aspirating to ensure that contact with a vessel had not occurred, between 6 and 10 cc of HA was injected within the lateral prostate capsule and the neurovascular bundle. This was performed under continuous TRUS guidance to view and verify the new space created by the injection. The total injected amount was related to the need for systematically, creating a minimum of a 1.5 cm space between the prostate, seminal vesicles, apex, and neurovascular bundle throughout this length.

Step 4: Under TRUS guidance, the neurovascular and lateral prostatic catheters with the microMOSFET were placed. The absorbed dose was calculated as the difference of the dose values between the points separated by the HA. We calculated the value of this decrease using the Oncentra Prostate planning system of Nucletron and measured this value using TN-502-RDM MOSFET detectors, hereafter microMOSFETs, manufactured by Best Medical Canada (Ottawa, ON, Canada).

The dimensions of microMOSFET detectors enabled their insertion into needles used for HDR brachytherapy. The detector response in voltage was measured before and immediately after each exposure or at the completion of treatment. The voltage difference between these measurements was proportional to the absorbed dose, which was obtained by applying a pre-determined calibration factor (CF) in Gy/V. In the clinical setting, these detectors were inserted into additional needles to those used in the treatment, placed at both ends of the HA thickness, allowing measurement of the true dose delivered. After correct insertion of the tip of the needle, monitored by US, the detector was placed in the end. In these conditions, the position of the sensitive part of the microMOSFET is approximately to 8 mm from the tip of the needle. The measurement system included a mobile MOSFET Reader TN-RD-16 electrometer that can simultaneously measure up to five detectors.

In the same plane of the ultrasound image in which the two microMOSFET were placed, and in the two points marked by the needles within which the detectors were inserted for each patient, the absorbed dose was calculated with the treatment planning system (TPS) once the planning of the treatment had been completed. In this way, we could compare the value of the decrease in absorbed dose obtained by both the TPS and the microMOSFET.

The microMOSFET was calibrated and employed by a water phantom. The absorbed dose for calibration at the measurement was 1 Gy, the source-detector calibration distance was 3 cm, and the dwell time required for this absorbed dose value at the calibration point was determined using the Oncentra Prostate planning system. The CF for twenty microMOSFET detectors was obtained, taking five consecutive measurements for each, to calculate its reproducibility.

All measurements and treatments were performed using Flexitron afterloader and the ^192^Ir V2r source (Nucletron/Elekta, Stockholm, Sweden).

Step 5: After the HDR treatment was completed, both catheters containing the microMOSFET detectors were removed for reading.

### 2.3. Uncertainties

We considered two groups of uncertainties, those associated with dose calculation by the TPS and those related to dose measurement by the microMOSFET detector [14,15]. For uncertainties of calculated absorbed dose, we considered source intensity (Air kerma strength, S_K_) and the interpolation associated with the TPS. For uncertainties of measured absorbed dose, uncertainty related to the global CF corresponding to each detector, we considered, in addition to the previous uncertainties, the source-detector calibration distance, the electrometer resolution, and the reproducibility.

### 2.4. Hyaluronic Acid

HA is a polysaccharide normally found in human tissues as a component of connective tissue and it has an absorption coefficient equal to that of water. Normally, it plays a vital role in the skin and in the synovial fluid of the joints. It is normally degradable by the normal enzymatic system in a relatively short time. However, to make it last for months when used for the treatment of skin wrinkles and osteoarthritis, the compound is modified, making it stable for a duration close to 1 year before it is reabsorbed by the body [16,17]. Only one type of HA is used in this trial.

## 3. Results

This is the first report in the medical literature using HA as a neurovascular bundle protector in patients with prostate cancer treated with only HDR interstitial brachytherapy as monotherapy in one fraction of 22 Gy.

### 3.1. Patient Characteristics

Our decision for neurovascular bundle preservation was considered appropriate in young men at the time of diagnosis and more interested in preserving sexual function, when biopsy examination demonstrated and confirmed imaging findings of organ-confined cancer with no suggestion of extracapsular extension to the neurovascular bundle. The biopsies and images were also evaluated by experienced pathologists and radiologists.

### 3.2. Radiologic Studies

Figure 1 and Figure 2 correspond to a magnetic resonance image (MRI) before and after HA injection demonstrating the newly created space between the prostate capsule and the neurovascular bundle.

### 3.3. MicroMOSFET Analysis

The mean values of the decrease in absorbed doses due to the HA thickness for three patients are shown in Table 1. TPS dose calculation and microMOSFET detector dose measurement are expressed in Gy and percentage. HA thicknesses were measured in the ultrasound image and are expressed in mm. All uncertainties correspond to a coverage factor k = 1.

The values (in percent) of the uncertainties that contribute to the TPS dose calculation are 1.5% for S_k_ and 2.6% for TPS interpolation, both taken from [18]. We estimated the uncertainty related to dose calculation as the mean quadratic root of the two previous uncertainties considered, which is 3.0%. We followed the same procedure to calculate the uncertainty of measurement of the absorbed dose, related to the global detector CF, which is 4.3%, but also considered other sources of uncertainty such as source-microMOSFET distance, 2.0%, resolution of electrometer, 0.009%, and reproducibility measured for twenty detectors, 2.3%. We considered an imprecision in the detector position of 0.5 mm to calculate the uncertainty in the calibration associated with source-detector distance. Considering the relative standard deviation (RSD) of five measurements for each detector, the reproducibility was calculated as the mean relative standard deviation for twenty detectors. RSD is defined as the ratio of the standard deviation (σ) to the absolute value of mean (|µ|). The uncertainty HA thicknesses are given by the precision of the measurement software of the ultrasound system. All uncertainties of the mean measurements for a coverage factor k = 1.

### 3.4. Tolerance

We have not seen side effects related to the injection or the compound itself. There has been no toxicity detected in the fat tissue neurovascular bundle or in rectal function. Patients have not complained of pain, tenesmus, pelvis pressure, or sensation of rectal filling. During the posttreatment follow-up evaluation, no patients complained of discomfort that could be attributed to the HA injection.

Because this work was started very recently, the results will be reported separately in a different publication after enough cases have undergone statistical analysis.

## 4. Discussion

The relationship between the probability of tumor control and dose has long been established [19,20]. The principal aim of conformal brachytherapy is to reduce irradiation to organs at risk, which enables a higher dose to the target volume. However, the increase in radiation dose to the prostate also has an impact of higher probability of toxicity in the neurovascular bundle [21]. The post-radiation injury mechanism is related to the damage of the vascular endothelium of the small vessels and arterioles and subsequent development of edema and fibrosis in nervous, vascular, and muscular tissue [22]. The consequence is the increased fragility of the neurovascular bundle and their tendency to erectile dysfunction.

The technique developed appears to be effective in keeping the neurovascular bundle away from the ionizing radiation. Hyaluronic acid was never injected in the prostatic stroma but was placed into periprostatic fat and absorbed locally by the lipocytes, and it did not leak into the prostatic stroma. For these reasons, we felt it was safe to use.

When a protection in the form of an HA layer is placed, the reduction effect at the maximum dose is between 46% and 54% (calculated values), which means the method for protection is highly recommended. The values of the absorbed dose calculated in this project have been compared with the ones given by the treatment planning system. These calculated values are similar to those given by the treatment planning system (56 ± 2), as can be seen in Table 1. This may be caused by the multiple approximations we have taken into account and because of the layout of the needles.

When the treatment is external beam radiotherapy or brachytherapy, the ionizing radiation can damage the neurovascular bundle. To avoid morbidity, it is recommended to reduce dose and minimize the volume of neurovascular bundle receiving high doses, but these procedures increase the likelihood of tumor persistence, thereby compromising oncologic control.

Several authors [23,24,25] have tested the application of the same concept to critical structures (rectum) and have presented a correlation of increasing rectal mucosal dose with increased rectal toxicity. In their multivariate analysis, dose was the only significant factor associated with Grade 2–3 of rectal toxicity. They concluded that chronic rectal toxicity is sequelae of high dose conformal treatment of prostate cancer. Appropriate shielding of the rectal mucosa by limiting the dose and to minimize the volume of rectum receiving high doses is required to avoid a high incidence of these complications.

In the same way, in our previous study [13], patients with low- and intermediate-risk prostate cancer were enrolled in a randomized clinical trial.

■The first group received brachytherapy alone with I-125■The second group received brachytherapy alone with I-125 and rectal protection with HA. HA was injected into perirectal fat to increase the distance between the prostate and the anterior rectal wall.

Of the non-hyaluronic group (first group), 4 out of 33 patients (12%) were suffering from intermittent rectal bleeding, and in these 4 patients, significant mucosal damage could be found. The total percentage of visible damage was 36% of the group. No patients with bleeding were observed in the group with rectal protection (HA), and only two patients presented with minimal objective mucosal damage at the endoscopy.

According to our data, the increase in distance between the rectum and posterior prostatic capsule created by the perirectal injection of HA is enough to provide a significant radiation dose and toxicity reduction.

We think that there is also a correlation between increase in dose and fragility of the neurovascular bundle and tendency to erectile dysfunction. The increase in distance between the prostatic capsule and neurovascular bundle created by the injection of HA is enough to provide a significant radiation dose reduction from HDR brachytherapy, and we believe many of the side effects can be either reduced or eliminated. No toxicity was produced by the HA or its injection.

Longer follow-up is needed to determine if there is a benefit from these planning exercises.

We are currently conducting a new study in which we will increase the number of measurements and thus be able to evaluate the results with better statistics.

## 5. Conclusions

When HA is injected (under TRUS guidance) transperineally into the lateral aspect of the prostate fat and the neurovascular bundle, creating a minimum of a 1.5 cm space, one can demonstrate the effect by MRI. This newly created space decreases the absorbed dose in the neurovascular bundle, calculated with the TPS and measured by microMOSFET due to the thickness of HA.

Longer follow-up is necessary to demonstrate the real gains associated with decreased neurovascular bundle doses.

## Figures and Tables

**Figure 1 biomedicines-10-00959-f001:**
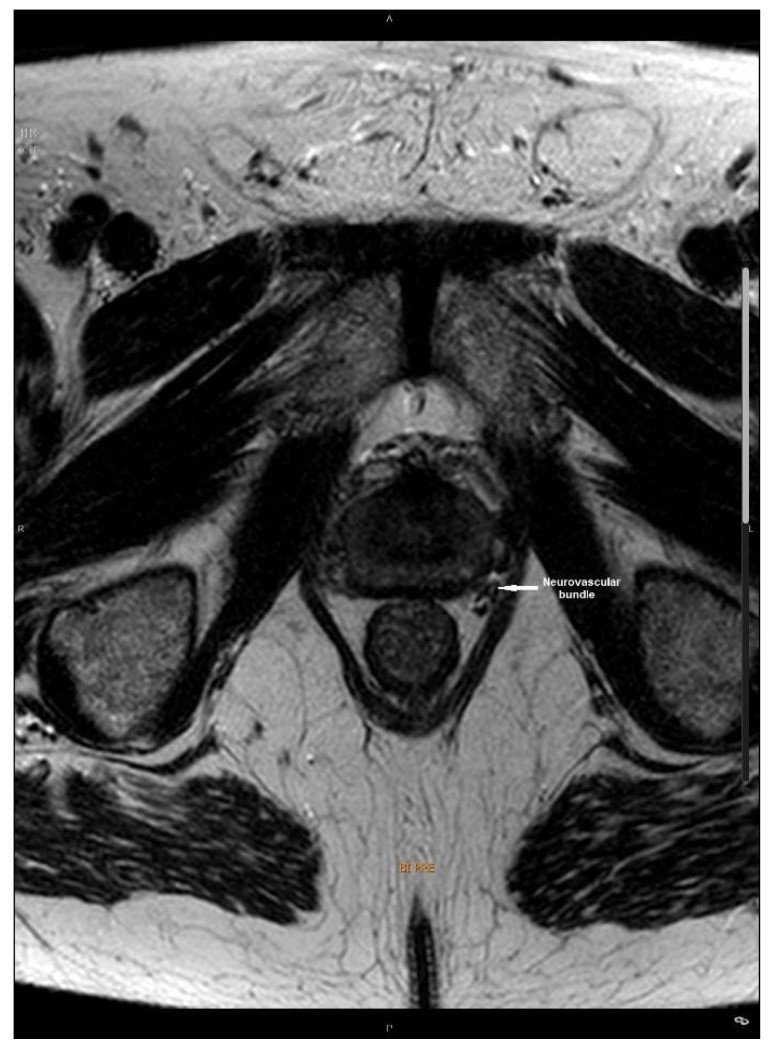
Magnetic resonance image identifying the neurovascular bundle.

**Figure 2 biomedicines-10-00959-f002:**
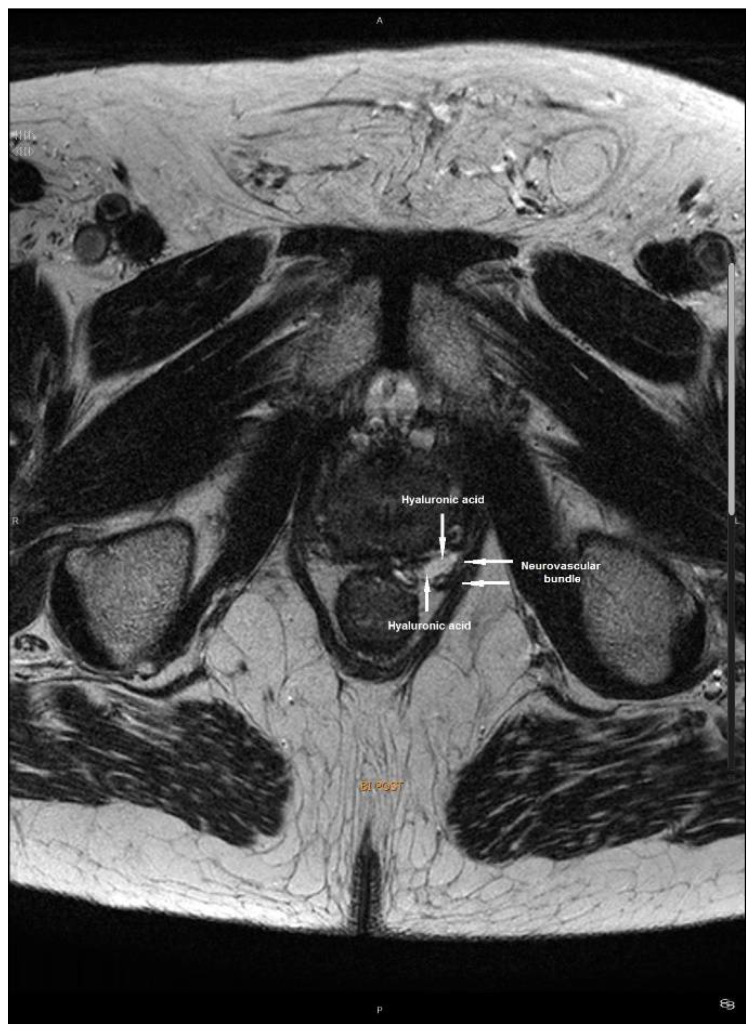
Magnetic resonance image demonstrating the additional space between the prostate capsule and neurovascular bundle.

**Table 1 biomedicines-10-00959-t001:** The table shows the mean values of the decrease in the absorbed dose, due to the thickness of HA, in the neurovascular bundle for each of the patients studied. The value of the decrease in absorbed dose calculated by treatment planning system (TPS) appears in the second (Gy) and the third (%) column, the value the decrease in absorbed dose measured by microMOSFET detector appears in the fourth (Gy) and the fifth (%) column, and HA thickness measured in the ultrasound (US) image appears in the sixth (mm) column.

Patient	Dose Decrease Due to HA				HA Thickness
#	TPS		microMOSFET		US (mm)
	(Gy)	(%)	(Gy)	(%)	
1	6.3 ± 0.4	51 ± 4	4.1 ± 0.5	42 ± 5	8.4 ± 0.1
2	4.3 ± 0.2	58 ± 4	4.3 ± 0.4	53 ± 5	11.1 ± 0.1
3	8.4 ± 0.5	59 ± 4	7.3 ± 0.6	53 ± 5	9.9 ± 0.1
Mean	6.3 ± 1.2	56 ± 2	5.2 ± 1.0	50 ± 4	10 ± 1

## Data Availability

Not applicable.

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
