# Peer review of "Intraoperative Neurovascular Bundle Preservation with Hyaluronic Acid during Radical Brachytherapy for Localized Prostate Cancer: Technique and MicroMosfet In Vivo Dosimetry"

_biomedicines, 2022, doi:10.3390/biomedicines10050959_

Round 1
Reviewer 1 Report
Authors present a new technique for intraoperative neurovascular bundle preservation with hyaluronic acid (HA) during radical brachytherapy for localized prostate cancer. The authors need to be congratulated on the study, it covers a very relevant topic. Introduction is reasonable, stats and concomitant results are sound. The discussion is coherent and interprets the findings with current literature. However, and only recently, a study could the benefits of active surveillance (AS) for eligible patients (PMID: 35053530). In light of these results, how do you see the clinical value of radical brachytherapy, and how would you set your findings (the benefits through HA) in context to such positive data in AS? What patients do you think would benefit most from brachytherapy with HA, or AS? Where do you see the future of brachytherapy with and without HA?
Author Response
With long-term data brachytherapy demonstrates excellent biochemical control for patients with localized prostate cancer. The potential for a therapy that is equally efficient but less harmful than other interventions is especially attractive for patients with early prostate cancer [1-4].
The positive aspects of modern brachytherapy offer the prostate cancer patients practical and logistical advantages over surgery and external beam radiation therapy.
- The median hospital stay for our patients was 12h (6-8 hours of implantation); there are not other alternatives of treatment with less hospital stay.
- Brachytheray does not cause incontinence.
- The sexual, urinary and gastrointestinal morbidity is very low.
- Patients treated with brachytherapy and rectal protection with hyaluronic acid had significantly smaller incidence of mucosal damage at the proctoscopic examinations and no macroscopic rectal bleeding [5,6].
On the other hand
Focal prostate therapies could reasonably be expected to bring a number of advantages [7].
- They will overcome the limitations about ‘‘active surveillance,’’ when is not accepted psychologically by some patients.
- They may provide a reasonable answer to accusations of ‘‘overtreatment,’’ when the tumor involvement is limited.
- Focal therapy is associated with no toxicity to noncancerous tissue and sparing key structures, such as the neurovascular bundles and the external urinary sphincter.
- Another advantage of this approach is the possibilities of a re-treatment (brachytherapy, specific techniques of external irradiation or salvage surgery).
- Burri RJ, Ho AY, Forsythe K, Cesaretti JA, Stone NN, Stock RG. Young men have equivalent biochemical outcomes compared with older men after treatment with brachytherapy for prostate cancer. Int J Radiat Oncol Biol Phys 2010; 77: 1315–21.
- Khan MA, Han M, Partin AW, Epstein JI, Walsh PC. Long-term cancer control of radical prostatectomy in men younger than 50 years of age: update 2003. Urology 2003; 62: 86–9.
- Zelefsky MJ, Marion C, Fuks Z, Leibel SA. Improved. Improved biochemical disease-free survival of men younger than 60 years with prostate cancer treated with high dose conformal external beam radiotherapy. J Urol 2003;170: 1828–32.
- Yamada Y, Bhatia S, Zaider M et al. Favorable clinical outcomes of three-dimensional computer-optimized high-dose-rate prostate brachytherapy in the management of localized prostate cancer. Brachytherapy 2006; 5: 157–64.
- Prada PJ, Fernandez J, Martinez A, et al. Transperineal injection of hyaluronic acid in the anterior peri-rectal fat to decease rectal toxicity from radiation delivered with intensity modulated brachytherapy or EBRT for prostate cancer patients. Int J Oncol Biol Phys. 2007; 69 (1):95-102.
- Pedro J. Prada, Herminio Gonzalez, Consuelo Menendez, et al. Transperineal Injection of Hyaluronic Acid in the Anterior Peri-rectal Fat to Decrease Rectal Toxicity from Radiation Delivered with Low Dose Rate Brachytherapy for Prostate Cancer Patients. Brachytherapy. 2009; 8(2): 210-217.
- Pedro J. Prada M.D, Ph.D1., Juan Cardenal M.D1., Ana García Blanco M.D1., Jon Andreescu M.D.1, María Ferri M.D1. Javier Anchuelo M.D1., Ivan Diaz de Cerio M.D1., Nicolas Sierrasesumaga M.D1. , Andrés Vázquez Ph.D2., Maite Pacheco Ph.D2., Samuel Ruiz Arrebola Ph.D2. FOCAL HIGH-DOSE-RATE BRACHYTHERAPY FOR LOCALIZED PROSTATE CANCER: TOXICITY AND PRELIMINARY BIOCHEMICAL RESULTS. Strahlentherapie und Onkologie. DOI: 10.1007/s00066-019-01561-3
Reviewer 2 Report
• The Authors should replace in the “Abstract” the term “background” with the term “purpose/outcome”.
• The Authors should explain that the subject of the work is also dosimetry and not just technique: so, they should modify the title, adding the term “dosimetry”.
• The Authors should shift the description of sexual function (2.1, page 2) from “Material and Methods” to “ Introduction”.
• The Authors should specify in the “Study Model” (2.2, pages 2-3):
1. numbers and characteristics of patients entered into the study;
2. the conducting and observation period of the study;
3. how the neurovascular bundle is correctly identified on Magnetic Resonance image (MRI) before the procedure;
4. if Magnetic Resonance pre-implant imaging fused with the intraoperative TRUS image can be used to identify during the procedure the neurovascular bundle, with more confidence compared to TRUS guidance;
5. if instead Magnetic Resonance pre and post-implant imaging can be used only to assess the size of the new space create.
Overall the work should be improved; however the work is original for the innovation of the topic and of interest to the readers.
Author Response
- The Authors should replace in the “Abstract” the term “background” with the term “purpose/outcome”.
- The Authors should explain that the subject of the work is also dosimetry and not just technique: so, they should modify the title, adding the term “dosimetry”.
- The Authors should shift the description of sexual function (2.1, page 2) from “Material and Methods” to “ Introduction”.
Requested corrections have been made at manuscript.
The Authors should specify in the “Study Model” (2.2, pages 2-3):
- numbers and characteristics of patients entered into the study;
The patients entered into the study were patients with histologically proven adenocarcinoma of the prostate. Memorial Sloan Kettering group definition was used to classify patients into risk groups: low risk and intermediate low risk patients.
- the conducting and observation period of the study;
Because this work was started very recently, the results will be reported separately in a different publication after enough cases are performed for statistical analysis.
- how the neurovascular bundle is correctly identified on Magnetic Resonance image (MRI) before the procedure;
Our Department of Radiology used multiparametric MRI and they identified the vascular bundle.
- if Magnetic Resonance pre-implant imaging fused with the intraoperative TRUS image can be used to identify during the procedure the neurovascular bundle, with more confidence compared to TRUS guidance;
Our brachytherapy equipment allows the fusion of images, magnetic resonance with TRUS. This has been added to the manuscript.
- if instead Magnetic Resonance pre and post-implant imaging can be used only to assess the size of the new space create.
To assess the size of the new space create and identify the vascular bundle.